# Potent New Targets for Autophagy Enhancement to Delay Neuronal Ageing

**DOI:** 10.3390/cells12131753

**Published:** 2023-06-30

**Authors:** Janka Szinyákovics, Fanni Keresztes, Eszter Anna Kiss, Gergő Falcsik, Tibor Vellai, Tibor Kovács

**Affiliations:** 1Department of Genetics, Eötvös Loránd University (ELTE), H-1117 Budapest, Hungary; 2Doctoral School of Biology, Institute of Biology, ELTE Eötvös Loránd University (ELTE), Pázmány Péter sétány 1/C, H-1117 Budapest, Hungary; 3ELKH-ELTE Genetic Research Group, H-1117 Budapest, Hungary

**Keywords:** ageing, autophagy, *Drosophila*, neurodegeneration, Parkinson’s disease, Rab2, Rab7, Arl8

## Abstract

Autophagy is a lysosomal-dependent degradation process of eukaryotic cells responsible for breaking down unnecessary and damaged intracellular components. Autophagic activity gradually declines with age due to genetic control, and this change contributes to the accumulation of cellular damage at advanced ages, thereby causing cells to lose their functionality and viability. This could be particularly problematic in post-mitotic cells including neurons, the mass destruction of which leads to various neurodegenerative diseases. Here, we aim to uncover new regulatory points where autophagy could be specifically activated and test these potential drug targets in neurodegenerative disease models of *Drosophila melanogaster*. One possible way to activate autophagy is by enhancing autophagosome–lysosome fusion that creates the autolysosome in which the enzymatic degradation happens. The HOPS (homotypic fusion and protein sorting) and SNARE (Snap receptor) protein complexes regulate the fusion process. The HOPS complex forms a bridge between the lysosome and autophagosome with the assistance of small GTPase proteins. Thus, small GTPases are essential for autolysosome maturation, and among these proteins, Rab2 (Ras-associated binding 2), Rab7, and Arl8 (Arf-like 8) are required to degrade the autophagic cargo. For our experiments, we used *Drosophila melanogaster* as a model organism. Nerve-specific small GTPases were silenced and overexpressed. We examined the effects of these genetic interventions on lifespan, climbing ability, and autophagy. Finally, we also studied the activation of small GTPases in a Parkinson’s disease model. Our results revealed that GTP-locked, constitutively active Rab2 (Rab2-CA) and Arl8 (Arl8-CA) expression reduces the levels of the autophagic substrate p62/Ref(2)P in neurons, extends lifespan, and improves the climbing ability of animals during ageing. However, Rab7-CA expression dramatically shortens lifespan and inhibits autophagy. Rab2-CA expression also increases lifespan in a Parkinson’s disease model fly strain overexpressing human mutant (A53T) α-synuclein protein. Data provided by this study suggests that Rab2 and Arl8 serve as potential targets for autophagy enhancement in the *Drosophila* nervous system. In the future, it might be interesting to assess the effect of Rab2 and Arl8 coactivation on autophagy, and it would also be worthwhile to validate these findings in a mammalian model and human cell lines. Molecules that specifically inhibit Rab2 or Arl8 serve as potent drug candidates to modulate the activity of the autophagic process in treating neurodegenerative pathologies. In the future, it would be reasonable to investigate which GAP enzyme can inhibit Rab2 or Arl8 specifically, but not affect Rab7, with similar medical purposes.

## 1. Introduction

In eukaryotic cells, there are two major forms of protein degradation, the ubiquitin–proteosome system, which degrades proteins that are labeled by a multiubiquitin signal [1], and the lysosome-dependent autophagic process that breaks down damaged and unnecessary cytoplasmic constituents including macromolecules and organelles [2]. Depending on the mechanism by which the cargo is delivered into the lysosomal compartment, three main types of autophagy can be distinguished: microautophagy, chaperone-mediated autophagy (CMA), and macroautophagy [3].

During macroautophagy (hereafter referred to as autophagy), a double-membrane structure called phagophore, or isolation membrane, forms in the cytoplasm around the materials destined to be degraded. The initial step of this process is termed vesicle nucleation. With the closure of the phagophore membrane, a vesicle-like structure is generated, called autophagosome, which later fuses with a lysosome to create an autolysosome. Alternatively, the autophagosome first fuses with an endosome to produce an amphisome, which also fuses with a lysosome. During lysosomal degradation, the enclosed cargo is degraded by acidic lysosomal hydrolases [4]. Because damaged macromolecules and organelles interfere with normal cellular functions, autophagy has a crucial cytoprotective role in eliminating such components. In addition, it also has important roles in macromolecule recycling, providing energy for cellular processes, and maintaining cellular homeostasis [5].

The hallmarks of ageing are the progressive deterioration of tissues and organs which causes the loss of physiological function and increased mortality rates. Nowadays, we can witness the increasing lifespan in developed societies, raising new challenges such as rising social and medical costs and finding treatment for age-related diseases. Experiments aiming for a better understanding of the mechanisms during ageing are vital for improving the health and quality of life of elderly populations [6]. The first evidence of autophagy playing a vital role in ageing was an experiment with *Caenorhabditis elegans,* which found that autophagy genes are essential for the lifespan extension of *daf-2* long-lived worms [7]. It was demonstrated in several genetic models that the capacity of autophagy declines with age, leading to a significant accumulation of cellular damage at advanced ages [8,9]. Thus, the process plays an important role in slowing the cell’s age rate [10,11]. For example, in the nematode *Caenorhabditis elegans*, loss-of-function mutations in autophagy genes (*Atg*), including *Atg1*/*unc-51* (uncoordinated), *Atg18/atg18,* and *Atg6*/*bec-1* (beclin 1 homologue), cause progeria and lifespan shortening [7,8,9,10,11,12]. Furthermore, enhancing autophagy by overexpressing *Atg8* or *Atg5* genes extends the lifespan in the fruit fly *Drosophila melanogaster* and the mouse *Mus musculus* [13,14]. A decrease in the autophagy capacity is particularly significant for post-mitotic cells such as neurons, which lose their ability to divide, so it is impossible to dilute damaged constituents and reproduce the lost cells by cell division [15]. The decline in autophagic degradation often leads to the incidence of various neurodegenerative disorders [16,17].

Autophagy is regulated by several signaling pathways whose lifetime-dependent changes in activity can also affect both autophagy capacity and lifespan [9]. For example, autophagy is downstream of the TGF-β (transforming growth factor-beta) signaling pathway, which is known to be a potential catalyst for cellular senescence [18]. TGF-β mutant animals (such as *daf-2*) defective in the pathway display up to a twofold increase in lifespan [19]. In *C. elegans*, the FoxO-like transcription factor DAF-16 links the insulin/IGF-1 and TGF-β pathways to regulate reproductive growth [19]. Ageing is regulated through the TGF-β pathway via the translocation of cytosolic DAF-16 to the nucleus. Due to crosstalk between signaling pathways, the JNK (c-Jun N-terminal kinase) signaling pathway is also a regulator of FoxO and, through it, autophagy [20]. Furthermore, many molecules in the human body affect lifespan by changing their amount. Many of these molecules also play an autophagy regulatory role. These substrates are absorbed through our diet, while other molecules that affect our metabolism are produced inside our tissues. Spermidine is the most abundant polyamine in the human body, and its concentration decreases continuously over the lifespan [21]. Sperm cells contain a high concentration of spermidine, which prevents cell ageing and allows the long-term survival of the germ cell line. Spermidine homeostasis is influenced by nutrition. The intake of spermidine is mediated by gut microflora, endogenous biosynthesis, intercellular degradation, and the efficiency of transport systems [22]. Similar to other caloric deprivation mimetics, spermidine activates protein deacetylation and autophagy [23]. Spermidine treatment increases survival rates in yeast, nematodes, fruit flies, and human immune cells and reduces age-related mortality in mice. In ageing flies, spermidine treatment improved memory, an effect that correlates with increased autophagy in neuronal tissue [24]. Several other nutritional substances have autophagy-activating and lifespan-influencing roles, such as urolithins (e.g., Uro-A), which activate autophagy in the nervous system and have neuroprotective effects. Its antiaging potency is achieved by autophagy degradation of damaged mitochondria and inhibiting ER stress [25]. Ursolic acid is a substrate that also can be absorbed with food and has both antiaging and autophagy-activating benefits [26,27]. Melatonin is a hormone produced by the pineal gland, released at night, which plays a role in regulating circadian rhythms. Melatonin’s beneficial role in lifespan has been demonstrated in *Drosophila* and mouse experiments, and it has a neuroprotective role [28,29]. It can activate autophagy through TOR, and insulin/IGF-1 pathways [30]. The above examples show that the level of autophagy during ageing is a complex mechanism influenced by multiple factors, which is not only a consequence of the expression levels of autophagy genes and the state of the signaling pathways that regulate them; the amount of antiaging substances intake in the diet, the state of their metabolism, or the autophagy activators produced in the tissues can all influence both lifespan and autophagy levels.

In neurons, autophagosomes are formed at the distal end of axons (right at the presynaptic membrane) and transported along microtubules by motor proteins, such as dyneins, to the soma (retrograde transport) [31,32]. To recruit dynein, autophagosomes need to fuse with late endosomes, resulting in structures called amphisomes [33]. During the transport to the soma, colocalization of the key autophagy proteins LC3 (Light chain 3B) and LAMP1 (Lysosomal-associated membrane protein 1) occurs, and the acidification of autophagic structures is also evident, suggesting the fusion of amphisomes with lysosomes [34]. Taken together, these observations indicate that in neurons, autophagosomes undergo a spatially defined maturation process, which requires endocytosis and retrograde transport [34].

Conditional ablation of the core autophagy genes *Atg5* and *Atg7* in the nervous system of mice impairs neuronal function and causes progressive neurodegeneration [35]. Knockout studies have demonstrated a crucial role for autophagy in embryonic neurogenesis, and mutations in autophagy genes lead to developmental delay, cognitive impairment, and functional deficits in childhood [36]. Additionally, the neuronal development is also influenced by autophagy, for instance, neuronal plasticities [35] or the regulation of synaptic vesicle formation [36,37]. Autophagy can delay the ageing process commonly associated with its ability to eliminate damaged proteins. During ageing, the progressive decline of autophagy can generate protein accumulation in neurons, causing neurodegenerative diseases. This emphasizes the importance of autophagy in nondividing tissues such as neurons [38]. The accumulation of damaged DNA also accompanies ageing. Several disorders, including Werner and Bloom syndrome, are characterized by DNA repair enzyme deficiencies leading to accelerated ageing. Interestingly, despite being a mainly cytoplasmic process, autophagy also impacts genomic integrity and DNA repair [35]. Autophagy dysfunction is involved in the pathology of several neurodegenerative conditions [39]. In Parkinson’s disease (PD), the mutant α-synuclein, which CMA normally breaks down, strongly binds to the LAMP2A (lysosome-associated membrane protein 2A) transporter on the lysosome, and therefore it is unable to translocate into the lysosome [40]. It also inhibits the translocation of other proteins into the lysosomal lumen, thereby increasing the accumulation of defective cytoplasmic proteins in the cytosol [40]. Furthermore, mutations in proteins that are involved in PD, including Parkin, PINK1 (PTEN-induced kinase 1), and PARK7 (Parkinson’s disease protein 7), also affect autophagic degradation. In case of mitochondrial damage, these proteins can label the outer mitochondrial membrane proteins with ubiquitins. Thus, they degrade defective organelles by a selective autophagic process called mitophagy. However, in the absence of these proteins, mitophagy is blocked, making the cell more sensitive to oxidative stress. Reactive oxygen species (ROS) released into the cytoplasm eventually lead to neuronal cell loss. Dopaminergic neurons in a specific area of the midbrain, substantia nigra pars compacta, are particularly affected by this process [41]. 

Currently, neurodegenerative diseases are considered as fatal human pathologies. This problem creates a great demand for understanding the molecular basis of such diseases and finding new targets for their effective treatment. The risk of developing various neurodegenerative diseases gradually increases with age. Therefore, there is intensive research on small molecules (drug candidates) activating autophagy. Such autophagy inducers identified so far largely influence processes upstream of autophagy, often leading to undesired side effects. Most of these inducers target TOR (target of rapamycin) kinase, which integrates the cellular information on nutrient and growth factor availability [42,43]. In yeast, there are two TOR paralogs, TOR1 and TOR2, while in higher eukaryotes, there is one TOR protein that participates in two complexes, TORC1 (TOR complex 1) and TORC2 (TOR complex 2) [44]. The function of these TOR paralogs and complexes differs from each other. TOR1 and TORC1 are associated with autophagy, cell growth, protein synthesis, cellular metabolism, and cell cycle regulation, while TOR2 and TORC2 regulate the cytoskeletal system and lipid synthesis. Thus, TOR regulates various cellular processes in addition to autophagy, so manipulating its activity can lead to severe side effects.

During vesicle nucleation, the VPS34 (Vacuolar protein sorting 34) kinase complex marks vesicles of various origins with PI3P (Phosphatidylinositol 3-phosphate), thereby linking these structures to the autophagic pathway [45]. An antagonist of this step is the MTMR14 (myotubularin-related lipid phosphatase) protein, capable of dephosphorylating PI3P through its phosphatase activity, thereby inhibiting the initiation of autophagy [46]. We previously showed that two autophagy enhancers, AUTEN-67 (autophagy enhancer 67) and AUTEN-99, exert their autophagy-activating effect by inhibiting MTMR14 [47,48]. These drug candidates increase autophagic activity in cell cultures and various model organisms, including *Drosophila*, zebrafish, and mice [47,48]. In *Drosophila*, both AUTEN-67 and -99 can promote longevity and improve the ability of aged animals to climb up on the wall of glass vials. AUTEN treatment reduces the ageing deformities of the *Drosophila* indirect flying muscle and improves the animal’s flying ability [49]. Moreover, these AUTEN molecules significantly lower the amount of toxic protein aggregates found in *Drosophila* models of PD and Huntington’s disease (HD), thereby increasing the survival of affected neurons [48,50].

AUTEN-67 and -99 activate the early stages of the autophagic process, phagophore, and autophagosome formation, at which the PI3K (type III PtdIns3) kinase complex functions. Some studies suggest that inadequate degradation of the autophagosomal content may contribute to the formation and accumulation of toxic proteins causing neurodegenerative processes [51]. Our previous study demonstrated that autophagic capacity gradually decreased in the brains of flies during ageing. In aged animals, decreased levels of early markers of autophagy, e.g., FYVE (Fab-1, YGL023, Vps27, and EEA1 domain)- and Atg5-labeled structures, indicate that fewer autophagosomes are formed, but the increased amount of Atg8a-positive structures and accumulation of lipid-conjugated Atg8a (Atg8a-II) show that autophagy is also impaired after autophagosome formation (when the structure fuses with a lysosome or the autolysosomal content is digested enzymatically) [52]. Thus, autophagy gradually declines with age in brain neurons due to the cumulative effect of suppressed autophagosome formation and compromised autolysosomal function. This prompted us to identify novel protein targets, the modulation of which can promote acidic degradation after autophagosome formation. One possible way is to improve autophagosome–lysosome fusion to enhance autolysosome formation. The fusion process is regulated by HOPS and SNARE complexes, and small GTPase proteins [29]. Small GTPases have key functions in membrane trafficking, and are involved in membrane docking, recruitment of other proteins, and membrane fusion [53]. As a molecular switch, the GTP/GDP-binding of small GTPases strongly influences the binding of effectors to the autophagic membrane. Whether Rab binds to GTP or GDP, it is dependent on enzymes. The GEFs (guanine nucleotide exchange factors) promote GTP-binding and conversion of small GTPase into an activated, membrane-bound state [54]. Small GTPase is inactivated by GAP (GTPase-activating proteins) enzymes, which support the hydrolysis of GTP to GDP, resulting in an inactivated, cytosolic form of Rab. Rab2, Arl8, and Rab7 proteins are important in lysosomal degradation pathways [55,56,57]. Studies on yeast and *Drosophila* reported that defective Rab7, Arl8, or Rab2 cause the accumulation of autophagosomes [55,58].

Three small GTPases, Rab2, Rab7, and Arl8, have a prominent and evolutionarily conserved role in regulating lysosomal degradation. These proteins differ in membrane affinities and function in acidic degradation.

Rab2 is highly conserved among *C. elegans*, *D. melanogaster*, and humans [55]. It is located on late endosomes, Golgi vesicles, and lysosomes [59], and a study found that in *Drosophila* muscle cells, the protein can bind to the autophagosomal membrane, where it binds to the HOPS complex and enhances the fusion of autophagosomes with lysosomes [60]. Progranulin (PGRN) plays an important role in regulating the lysosome. In its absence, lysosomal abnormalities, LC3-II (Atg8-II), and p62 accumulation can be observed. In HEK293T cells, PGRN can physically interact with Rab2, thereby playing an important role in autophagosome–lysosome fusion [61]. Interaction between Rab2 and HOPS is present in *Drosophila* [62] and in cells where human Rab2A promotes an invasive program of breast cancers [63]. Rab2 also has an important role in autophagosome and endosome maturation, is vital for normal lysosomal function, and regulates retrograde and anterograde transports between the Golgi and endoplasmic reticulum [59,64]. In a mouse model, Rab2 was shown to activate ULK1/Atg1; hence, it regulates autophagy via the induction of the ULK1 kinase complex [55].

Rab7 is located in autophagosomal and late endosomal membranes [57], necessary for the autophagic vesicles’ transport along microtubules and the maturation of autophagosomes and endosomes [65,66]. The protein is involved in various cellular processes, including autophagy, apoptosis, phagocytosis, and cytoskeleton organization. However, unlike in yeast and *Drosophila*, Rab7 is required for autolysosome maturation (Rab7B KO cells have lower levels of autolysosome cathepsin B) rather than autophagosome–lysosome fusion. The divergence in mammalian models might be due to multiple Rab7 orthologs, which could have partially redundant functions [67,68].Similar to Rab2, Rab7 also has specific autophagy binding partners, such as Rubicon. Rubicon can only bind to the autophagosome in complex with Rab7. Thus, Rab7 is also a negative regulator of autophagy as a member of the complex with Rubicon [69]. In neurons, Rab7 functions during axonal retrograde transport, neuronal migration, trafficking, and signaling of the neurotrophin receptor [70].

Arl8 is a member of the Arf family of small GTPase proteins, conserved across many eukaryotic taxa, and can be found on the lysosomal membrane [71]. In human and *Drosophila* cells, the protein is essential for the optimal distribution of lysosomes [72]. Arl8 overexpression causes a peripheral distribution of lysosomes. Moreover, Arl8 plays a key role in the trafficking of components taken up by endocytosis to the lysosome, and the axonal transport of synaptic vesicle precursors [72]. In *Drosophila*, it is vital for axonal transport and endosomal function. Besides positioning lysosomes, Arl8 mediates endosome–lysosome and autophagosome–lysosome fusions by interacting with the HOPS complex [56]. Furthermore, *Drosophila* experiments have demonstrated that Rab2 and Arl8 are involved in the bidirectional axonal transport of amphisomes [67]. In macrophage cell cultures, Arl8 is required to form tubular lysosomes, which have significantly higher motility than punctate lysosomes [73]. 

Several research groups are investigating the role of small GTPases in neurodegenerative diseases. *Drosophila* Rab7 deficiency causes neurodegeneration [74]. Furthermore, overexpression of the wild-type form of Rab7 rescues the neuronal cell death phenotype in α-synuclein (A53T) mutant rats [75]. Overexpression of the wild-type form of Arl8 can further impair A53T-induced toxicity [76]. In contrast, the constitutively active form of Arl8 could rescue Aβ toxicity. Arl8 appears to be a susceptible point in neuronal transport processes in neurodegenerative diseases. Overexpression of a human mutant form of LRRK-2 (leucine-rich repeat kinase 2) involved in Parkinson’s disease causes accumulation of Arl8- and Rab3-positive vesicles at the terminal end of axons, leading to defects in axonal transport [77,78]. Arl8 also interferes with the lysosomal transport of axons in neurodegenerative lysosomal storage disorder Niemann–Pick disease type C (NPC).

In this study, we examined the effect of overexpressing constitutively (GTP-bound) active Rab2, Rab7, and Arl8 (Rab2-CA, Rab7-CA, and Arl8-CA) on the nervous system of *Drosophila melanogaster.* We aimed to test whether the activation of small GTPase proteins has a beneficial effect in this model system. We also inspected Rab2-CA and Arl8-CA overexpression in a PD model.

## 2. Materials and Methods

### 2.1. Animals

All tests were performed on a standard corn flour, sugar, and agar medium at 29 °C. We used the UAS-Gal4 system to express y^1^ w^*^; P{w[+mC] = UASp-YFP.Rab2.Q65L}02 (constitutively active—UAS-Rab2-CA; BDSC:9760), y^1^ w^*^; P{w[+mC] = UASp-YFP.Rab7.Q67L}19 (UAS-Rab7-CA^#24103^—BDSC:24103), y^1^ w[*]; P{w[+mC] = UASp-YFP.Rab7.Q67L}awd [14]/TM3, Sb^1^ (Rab7-CA^#9779^—BDSC:9779), P{w[+mC] = UASp-YFP.Rab7.Q67L}7, y^1^ w* (Rab7-CA^#50785^—BDSC:50785), w^*^; P{w[+mC] = UAS-Rab7.Q67L}2 (Rab7-CA^#42707^—BDSC:42707) specifically in dopaminergic and serotonergic neurons (Ddc-Gal4; BDSC:7010). The following *Drosophila* stocks were used for gene silencing: 40D-UAS (as an RNAi control—VDRC:60101), w^1118^; P{GD11800}v40337 (UAS-Rab7-RNAi—VDRC: 40337), w^*^, Arl8[NIG.7891R (UAS-Arl8-RNAi—VDRC: 7891R-2), P{KK107630}VIE-260B (UAS-Rab2-RNAi—VDRC: v105358). We also used y^1^ w^*^; P{w[+mC] = UAST-YFP.Rab2}l(3)neo38^02^/TM3, Sb^1^ (UAS-Rab2-WT—BDSC:23246) flies to compare the overexpression of wild-type Rab2 to the overexpression of a constitutively active form of Rab2 (Rab2 CA), and y^*^ w^*^; P{UAS-Luc.VALIUM20}attP40 (UAS-Luc—gift of Tamás Lukácsovich [52]) to compensate 2 Gal4 sequences in a PD model w^*^; P{UAS-Hsap\SNCA.A53T} [15]. (UAS-A53T—BDSC:8148). y^*^ w^1118^; 3xmCherry-Atg8a^4–14^ (mCherry-Atg8a) and w^*^; UASpArl8.Q67L (Arl8-CA) flies stocks were a gift from Gábor Juhász [79]. We also used w^1118^ isogenic flies as a control (w—BDSC: 5905).

### 2.2. Lifespan Assays

For this experiment, 150 animals per genotype (75 male and 75 female) were used. During the lifespan assay, animals were kept at 29 °C. We measured the number of dead flies daily and transferred the living animals into new tubes every second day.

### 2.3. Climbing Assay

The climbing tests were carried out as described in the publication [52]. In the climbing tests, we examined the sexes separately because there could be a significant difference between the climbing ability of males and females. We tested 10 animals in a single tube, with several parallel experiments per genotype. The average climbing ability of each point on the boxplots is shown for these parallel measurements. After sorting animals into climbing tubes with anesthesia, we provided them 1.5 h to regenerate at 29 °C before the test. In this experiment, the negative geotaxis reflex of the animals was examined [48]. We used 25 cm long glass vials to monitor the ability of animals to climb on the wall of these test tubes. We gave the animals 60 s to climb (and counted the animals at 20, 40, and 60 s that reached the bottom (6 cm) and the top (21 cm) line and we repeated the measurement 3 times with 30 min breaks between them. We measured the climbing ability at 7, 14, 21, and 28 days.

### 2.4. Microscopy

Fluorescent images were captured with a Zeiss Axioimager Z1 upright microscope (with objectives Plan-NeoFluar 10 × 0.3 NA, Plan-NeoFluar 40 × 0.75 NA and Plan-Apochromat 63 × 1.4 NA) equipped with ApoTome2. We used AxioVision 4.82 and ImageJ 1.52c software to examine and evaluate data.

### 2.5. Western Blotting

Protein samples of flies stem from female adult heads. For protein samples, solutions containing 2 mg protein were prepared using 15 female heads. The heads were homogenized in 48 μL lysis buffer and 48 μL mercaptoethanol-containing lysis buffer (BioRad, 1610737, Hercules, CA, USA). The lysis buffer contained the following components: 0.5% Tween 20, 25 mM Tris pH 7.5, 150 mM NaCl, 2 mM EDTA, 20 mM Nicotinamide, and 6.51 mL MQ water. Then, 15 μL samples were loaded on a 4–20% SDS-PAGE and blotted onto Immobilon P PVDF membrane (Millipore, IPVH00010, Burlington, MA, USA). Next, we performed blocking with 0.5% blocking reagent (Roche, 1096176, Basel, Switzerland) in PBST (PBS containing 0.1% Tween 20). Membranes were probed with the following antibodies: anti-Ref(2)P/p62 (rabbit, 1:2500) [80], alpha-Tub84B (mouse, 1:2500, Sigma, T6199, St. Louis, MI, USA), anti-Atg8a (rabbit, 1:2500) [81], anti-α-synuclein (mouse, 1:1000, BD Biosciences, 610787, Franklin Lakes, NJ, USA), anti-Arl8 (rabbit, 1:1000, DSHB, AB 2618258), anti-Rab7 (mouse, 1:1000, DSHB, AB 2722471), anti-Rab2 (rabbit, 1:200, Santa Cruz Biotechnology, Inc., FL-212-sc:28567, Dallas, TX, USA) [82]. The secondary antibodies, anti-rabbit IgG alkaline phosphatase (1:1000, Sigma, A3687), anti-mouse IgG alkaline phosphatase (1:1000, Sigma, A5153), were visualized by NBT-BCIP solution (1:50, Sigma, 72091). Western blot experiments were repeated at least three times. We evaluated the Western blot results using ImageJ software. 

### 2.6. Immunohistochemistry

Female *Drosophila* heads (mouthparts were removed previously) were fixed in 4% formaldehyde for 45 min at room temperature, then we washed the samples 3 times for 30 min in PBST solution. After the last washing, the brains were dissected and blocked for 1 h in fetal bovine serum (solved in PBST). Primary antibody labeling (in blocking solution) was performed for 2 days at 4 °C with the following antibodies: antiubiquitin (mouse, 1:500, Sigma-Aldrich, ST1200) and anti-α-synuclein (1:1000, BD Biosciences, 610787). On the third day, the brains were washed 3 times for 20 min in PBST solution and incubated for an additional 20 min in blocking solution. Secondary antibody labeling was performed for 2 h at room temperature; secondary antibody: Alexa Fluor 488 goat anti-mouse lgG (1:500, Invitrogen, A11001, Waltham, MA, USA). Samples were washed 2 times in PBST and once in PBS. For microscopy, samples were covered with a 1:4 PBS: glycerol solution containing Hoechst (nucleus stain).

### 2.7. PCR and Quantitative PCR

Isolation of mRNA from female heads (for RNA isolation, 10 female heads were used for each sample) was performed according to the Direct-zol™ RNA MiniPrep kit (Zymo Research, R2050, Irvine, CA, USA) protocol, then cDNA was generated by RevertAid RT Reverse Transcription Kit (Thermo Scientific, K1691, Waltham, MA, USA). Quantitative real-time PCR reactions were performed in a Roche LightCycler 96 Instrument (Roche Molecular Systems) with FastStat Essential DNS Green Master kit (Roche, 06924204011). Actin5C mRNA level was used as an inner control. Forward (F) and reverse ^®^ primers were as follows: *Rab2* F: 5′-TGT CGC TAG CCA GTC ATC AT-3′ and R: 5′-CGT GTG ATA GAT CTG AAA GCC TC-3′, *Rab7* F: 5′-GGT CAC AAT GCA GAT CTG GG-3′ and R: 5′-CGC CAG GAG TCG AGA TTC TT-3′, *Arl8* F: 5′-CGT CAA TGT TAT TGC ATC CGG-3′ and R: 5′-TGA GTC CGG TTT CAT CGA GA-3′ *actin5C* F: 5′-GGA TAC TCC CGA CAC AA-3′ and R: 5′-GAG CAG CAA CTT CGT CA-3′. To determine relative mRNA levels, the results were corrected for the variation in *actin5c* mRNA (internal control) levels. The data were calculated according to the following method [83].

### 2.8. Statistical Analysis

For statistical analysis of climbing assays, lifespan measurements (mean lifespan), and fluorescence microscopy, results were determined using R Studio (Version 3.4.3). Distribution of samples (normal or not) was tested with Lilliefors test. If it was normal, F-test was performed to compare variances. In cases when variances were equal, a two-sample t-test was used, otherwise t-test for unequal variances was applied. In case of non-normal distribution, a Mann–Whitney U-test was performed. We calculated Pearson’s coefficients by Image J 1.52c, for evaluating the colocalization of mCherry-Atg8a and YFP-Rab2 particles. The logrank (Mantel-Cox) method was used for lifespan curve statistics, calculated with the SPSS17.0 program.Kaplan-Meier curves were created by SPSS17.0 program.

## 3. Results

### 3.1. Results

#### 3.1.1. Small-GTPase Proteins Involved in the Lysosomal Degradation Pathway Are Required for Normal Longevity and Viability of Flies

In the *Drosophila* model system, some small GTPases are known to regulate cognitive and physiological abilities. For example, the overexpression of the small GTPase Rheb, which regulates TOR, results in elongated axon spines and cell bodies in the mushroom body. Increased levels of Rheb are also associated with memory impairment [84]. The lack of Ras-family Rin and Ric subfamily members shows neuronal morphology and survival abnormalities [85]. The photoreceptor cells of the compound eye derive from the nerve. The development of the R7 photoreceptor cell is regulated by Ras and Rap small GTPases [86]. Rho small GTPase is involved in the plasticity of the nervous system. It remodels actin-rich dendritic spines, in the absence of which, in invertebrates, learning and memory impairments occur [87]. 

In our experiments, we first examined whether the absence of small GTPases (similar to other small GTPases), which are essential in acid degradation, might cause damage to the nervous system that affects animal survival and behavior function. Our results show that the silencing of *Rab2, Rab7*, and *Arl8* inhibits autophagy, reduces the lifespan of animals, and impairs their locomotor ability (Figure 1 and Figure 2).

Dopaminergic neurons (DNs) produce the neurotransmitter dopamine, the loss of which can lead to the development of PD. In our first measurements, we aimed to investigate the role of small GTPases on the physiological and cognitive abilities of *Drosophila*. All three small GTPases (Arl8, Rab2, Rab7) were silenced by RNA interference (RNAi). In a previous study, we showed that autophagic activity is not constant in *Drosophila* adult brains, with some neurons being more sensitive to changes in autophagy levels. Increasing autophagy in dopaminergic and serotonergic neurons (via inhibition of EDTP) may result in lifespan extension and improved cognitive function [52]. Therefore, we used Ddc-Gal4 expressed exclusively in dopaminergic and serotonergic neurons [88].

We measured the efficiency of the RNA interference strains by quantitative PCR and Western blot analysis (Figure A1A,B). The used *Rab2*-RNAi construct could reduce transcript and protein levels, and *Arl8* and *Rab7* silencing only reduced protein levels (translational inhibition). Autophagy was inhibited upon small-GTPase silencing. Ref(2)P/p62 is a substrate protein for autophagy, the amount of which is inversely proportional to the efficiency of autophagy [89]. In a Western blot analysis, Ref(2)P levels were elevated during *Rab2*, *Rab7*, and *Arl8* silencing. Compared to the Atg8a-I and -II ratio in control samples, small GTPase silenced samples accumulated Atg8a-I soluble protein in aged samples. This result also suggests a failure in autophagy (Figure 1B and Figure A2A,A’). The ubiquitinated aggregates and cell organelles are also degraded by autophagy [90], which were labeled by immunohistochemistry and studied by fluorescence microscopy. Silencing of *Rab2*, *Rab7,* and *Arl8* increased the number of ubiquitin-positive structures in *Drosophila* adult brains (Figure 1C,C’ and Figure A2B,B’). Finally, we examined the abundance of mCherry-Atg8a-positive structures by fluorescence microscopy (which indicates all autophagic structures from the phagophore to autolysosome). mCherry-positive autophagic structures accumulated in these three cases of gene silencing (Figure 1D,D’ and Figure A2C,C’).

Then, we examined the effect of silencing *Rab2*, *Rab7,* and *Arl8* in the nervous system of *Drosophila.* Rab2-, Rab7-, and Arl8-RNAi in DNs shortened lifespan (Figure 2A,A’ and Figure A2D,D’). The *Drosophila* climbing assay can be used to investigate the fitness of animals and their response to a given stimulus. To measure the cognitive abilities of animals, it can be tested in a 25 cm long tube. By knocking down the tube, negative geotaxis (upward climbing) can be induced in the animals. From the speed of the climb, the response time of the animals to the stimulus can be inferred [48,50,52]. Silencing all three small-GTPases impaired the animals’ climbing ability (Figure 2B,B’). Our results suggest that small-GTPases are required for normal climbing ability and autophagy in *Drosophila* neurons. After this, we investigated the physiological and cell biological consequences of activating small-GTPase proteins.

#### 3.1.2. Rab2 and Arl8 Activation Extends Lifespan and Improves Climbing Ability

Following the silencing of small GTPase, we examined the effects of activating Rab2, Rab7, and Arl8 (overproduction of GTP-bound forms) on neuronal ageing (Figure 3). Our results show that activation of Rab2 and Arl8 enhances autophagy, whereas Rab7-CA inhibits acid degradation. Enhancement of autophagy may contribute to the increased lifespan and locomotor ability observed in animals. In contrast, Rab7-CA, which inhibits autophagy, is detrimental to the viability and motility of the animals (Figure 3). 

The small GTPase activation was achieved by neuronal (dopaminergic and serotonergic specific) overexpression of constitutively active, continuously GTP-bound mutants (hereafter referred to as Rab2-CA, Rab7-CA, and Arl8-CA) [56,82,91]. We examined the effects of Rab2-CA, Arl8-CA, and Rab7-CA on lifespan, and found that the overexpression of activated forms of Rab2 and Arl8 significantly increased lifespan. In contrast, Rab7-CA significantly reduced lifespan relative to control (Figure 3A and Figure A3A). In aged animals, Rab2-CA and Arl8-CA improved, while Rab7-CA decreased, the ability of animals to climb compared to control (Figure 3B). Therefore, activated Rab2 and Arl8 exert a beneficial effect in aged flies. We also compared the effects of overexpressing constitutively active or wild-type forms of the investigated proteins and observed that only Rab2-CA and Arl8-CA have beneficial impacts compared to the WT forms. Rab7-WT and -CA were both detrimental, but Rab7-WT animals reached a longer lifespan (compared to age-matched Rab7-CA) (Figure A3B).

We compared the autophagy levels in samples isolated from the heads of small GTPase-WT and -CA animals. In young animals, Rab7-CA and Rab7-WT increased Ref(2)P, suggesting inhibition of autophagy. Rab7-overexpressing animals (both WT and CA) did not reach 28 days of age, so at this time, only Rab2 and Arl8 activation was compared to overexpression of wild-type forms. In older animals, only Rab2-CA and Arl8-CA decreased Ref(2)P levels, in contrast to the overexpression of -WT forms where a slight increase was observed in the relative amount of Ref(2)P (Figure A4A–A’’’). Climbing ability was also impaired in these animals, suggesting that overexpression of Rab2-WT and Arl8-WT is detrimental in older animals (Figure A3B).

The adverse effects of Rab7-CA could result from either the existence of activated Rab7 or the harmful expression of the protein. To address this issue, we compared the expression levels of Rab2-CA and Rab7-CA. Rab7^#24103^ (the originally used construct) showed significantly higher expression levels than Rab2-CA (Figure A3C,C’). The expression of three additional Rab7-CA alleles, Rab7-CA^#9779^, Rab7-CA^#50785^, and Rab7-CA^#42707^, were also assessed. Their expression levels were significantly lower than Rab7-CA^#24103^ and Rab2-CA (Figure A3C,C’). Despite these data, the neuronal expression of Rab7-CA^#9779^ and Rab7-CA^#50785^ alleles harmed the climbing ability of transgenic animals (Figure A3D). These results suggest that Rab7 activation may have a detrimental effect on the climbing ability of animals.

We also tested the effect of small GTPase activation on autophagic activity. Ref(2)P and Atg8a labeling were performed on protein samples isolated from heads by Western blotting. We observed a decrease in Ref(2)P levels in response to Rab2 and Arl8 activation, whereas increased Ref(2)P levels were noticed in the case of Rab7-CA (Figure 3C and Figure A3E,E’). Arl8-CA and Rab2-CA also led to a decrease in Atg8a-II levels compared to Atg8a-I (soluble form). For autophagic vesicle labeling, we used an mCherry-Atg8a reporter, which showed decreased Rab2-CA and Arl8-CA levels, but an accumulation in Rab7-CA levels, in brain samples (Figure 3D–D” and Figure A3F). In the presence of continuously elevated autophagy, Atg8a can also be considered a substrate for acid degradation, so decreased mCherry labeling and lowered Ref(2)P levels suggest autophagy activation. We next monitored the colocalization of the Rab7-CA marker with mCherry-Atg8a-positive structures and found several cases when green and red vesicles remained nonoverlapped (Figure 3E). These results indicate defects in autophagosome/late endosome–lysosome fusion. A similar colocalization feature to Rab7-CA was not observed between YFP-Rab2-CA and mCherry-At8a (Figure A4B,B’).

Finally, antiubiquitin labeling was used to investigate the amount of ubiquitinated structures in *Drosophila* brain samples. Both Rab2-CA and Arl8-CA were able to reduce, whereas Rab7 activation caused a significant increase in, the amount of ubiquitin-positive structures (Figure 3F,F’). These findings suggest that Rab2-CA and Arl8-CA can enhance the autophagic process in the *Drosophila* nervous system, and benefit neuronal functions in aged flies. Moreover, increasing the GTP-bound form of Rab7 in neurons can interfere with viability, which may result from, at least in part, the inhibition of autophagy.

#### 3.1.3. Rab2 Activation Promotes Longevity in a Parkinson’s Disease Model

The lack of autophagy has been associated with a number of neurodegenerative (ND) diseases. In contrast, enhancing acid degradation can benefit the survival of dying ND neurons, prolong the lifespan, and delay the onset of disease-associated dysfunctional symptoms. In the previous chapter, the induction of active forms of Rab2 and Arl8 stimulated autophagy function and had beneficial effects on animals during ageing (Figure 3). Therefore, it was considered worthwhile to investigate the effects of Rab2-CA and Arl8-CA in Parkinson’s disease (PD) models (Figure 4). Our results show that the neuronal activity of both Rab2 and Arl8 can enhance autophagy, stimulate the degradation of the toxic protein (that causes PD), and improve locomotor abilities in aged animals (Figure 4).

In humans, PD can result from an autosomal dominant mutation (e.g., A53T) of *SCNA* gene encoding α-synuclein protein [92]. The mutant protein is more likely to produce fibrils (A53T also triggers Tau fibrillation) than the wild-type one, thereby promoting disease progression [93]. We used Ddc-Gal4 driver to express Rab2-CA and Arl8-CA in dopaminergic and serotonergic neurons with an A53T background. UAS-Luc; UAS-A53T was used to create a control fly model of PD. According to these data, Rab2-CA increased survival and climbing in transgenic animals at advanced ages compared to age-matched control (Luc, A53T) (Figure 4A,B and Figure A5A). This indicates that the UAS-Luc; UAS-A53T transgene negatively influences climbing ability and lifespan, thereby successfully generating the PD model. In Arl8-CA animals, we did not observe a significant change in lifespan compared to PD control but noticed an improvement in the climbing of old animals (Figure 4A’,B’ and Figure A5A’).

Next, we examined changes in autophagic activity during ageing in flies serving as a model for PD. In these flies, Rab2-CA or Rab2-WT were overexpressed. We found that Ref(2)P/p62 levels increase in aged animals compared to young ones (Figure 4C). This implies that autophagy becomes compromised during ageing in transgenic animals. Contrary to this, overexpression of Rab2-CA enhanced autophagic activity compared to Rab2 WT-overexpressing flies (Figure 4C). Thus, wild-type overexpression of Rab2 alone does not induce A53T degradation (toxic A53T protein degradation requires Rab2-CA expression). In animals overexpressing Rab2-CA, the amount of lipid-bound Atg8a was reduced, suggesting an efficient degradation of the inner autophagosomal membrane (Figure 4C and Figure A5C). Similar to Rab2-CA, Arl8 activation also enhanced autophagy in aged animals. The amounts of Ref(2)P and Atg8a-II proteins were reduced compared to control PD (Luc; A53T) (Figure 4C’ and Figure A5C’). Thus, both Rab2-CA and Arl8-CA can activate autophagy in neurons of A53T mutant flies.

Finally, we also studied changes in the amount of toxic A53T protein by fluorescence microscopy and Western blotting (Figure 4D,D’,E and Figure A5B,B’,E). Human α-synuclein-specific antibody was used for labeling. Confocal imaging allows the detection of punctate structures (aggregates), whereas conventional imaging is more favorable for determining the soluble protein amount [94]. Results showed that Rab2 and Arl8 activation could reduce soluble and aggregated A53T amounts (Figure 4D,D’ and Figure A5B,B’). A Western blot analysis led to similar results. The overexpression of both small-GTPases-CA reduced A53T levels (Figure 4E and Figure A5E). For Rab2-CA, proteins were separated into soluble and nonsoluble fractions by centrifugation (Figure A5D) [95]. In both cases, a decrease in A53T abundance was observed in response to Rab2-CA. In summary, activation of Rab2 and Arl8 in neurons of flies serving as a model of PD can activate autophagy, decrease toxic A53T levels, and improve survival and movement abilities.

## 4. Discussion

In the present study, we investigated the effects of neuronal-specific silencing of *Rab2*, *Rab7,* and *Arl8* on lifespan, climbing functions, and autophagic activity. Results we obtained show that silencing these small-GTPases harms animals’ survival and climbing. Furthermore, autophagy is inhibited in neurons overactivating these proteins (Figure 1, Figure 2 and Figure A2).

We revealed that overexpression of Rab2-CA and Arl8-CA each reduces the level of autophagy substrate Ref(2)P/p62 and ubiquitin proteins in dopaminergic and serotonergic neurons, and improves the movement of aged flies (Figure 3). The amounts of Ref(2)P and ubiquitin-positive structures were reversely altered in Rab7-CA samples compared to Rab2-CA and Arl8-CA. Thus, whereas the activation of Rab2 and Arl8 enhances autophagy, Rab7 impairs the degradation process. Furthermore, Rab7 activation appears to be detrimental to the animal’s climbing ability and viability (Figure 3 and Figure A3). The constitutively active form of Rab7 caused mCherry-Atg8a accumulation in the *Drosophila* brain. The colocalization of Rab7- and Atg8a-positive structures is impaired in Rab7-CA animals (Figure 3E). In HeLa cells, both Rab7 and Arl8 can bind to late endosomes [96]. The activation or inactivation of Rab7 affects the localization of Arl8 vesicles [97]. We hypothesized that Rab7 activation inhibits amphisome formation (amphisomal vesicles formed by the fusion of endosomes with autophagosomes). Autophagosomes forming at the end of axons undergo a retrograde transport into the soma. Prior to the transport, endosomes and autophagosomes fuse to create amphisomes. Finally, the autophagosomal cargo is degraded in the soma [31]. It can be assumed that autophagy inhibition and lifespan shortening in Rab7-CA animals are partly due to the lack of Rab7–Arl8 exchange on endosomes and inhibition of retrograde transport, respectively.

Overexpression of Rab2-CA in PD model animals (flies express the human A53T mutant α-synuclein) extends lifespan (Figure 4A). Both Rab2-CA and Arl8-CA improve climbing ability compared to PD control (Figure 4A–B’ and Figure A5A,A’). In addition, these hyperactivated small GTPases increase autophagy and degradation of toxic α-synuclein relative to the control group (Figure 4 and Figure A5). Cytoplasmic α-synuclein can be found in as soluble monomers, oligomers, and aggregates. The accumulation of α-synuclein aggregates is a characteristic feature of PD. However, soluble oligomers are toxic to the cell [98,99]. The degradation of α-synuclein dissolved in the cytoplasm was more significant than the elimination of α-synuclein-positive aggregates. In PD model animals expressing A53T mutation, only Rab2 and Arl8 activation was tested, and it was found to have positive effects on wild-type animals. Rab7-CA negatively affects viability and autophagy (Figure 3). In HEK293 cells, overexpressing Rab7 reduces the amount of mutant α-synuclein [100]. Furthermore, Rab7 overexpression rescues the climbing ability defective phenotype in a fly model of PD [100]. Although overexpression of Rab7-WT was detrimental to the climbing ability of the tested animals (Figure A3B), overexpression of Rab2-CA led to the opposite results, as only the latter was able to increase climbing ability in the affected animals. Similar results were also obtained when comparing Arl8-WT and Arl8-CA (Figure A3B). In this PD model, Rab-CA and Arl8-CA could trigger autophagy (Figure 4C,C’). It might be interesting to compare the effects of Rab7-CA and Rab7-WT overexpression between wild-type and PD model animals.

Although these small GTPases act as regulators of autophagy, they also function independently of acidic degradation. For example, Arl8 plays a role in controlling microtubule and vesicle transport [101]. The protein promotes the axonal transport of presynaptic cargo proteins, thereby preventing premature aggregation [102]. Rab2 is a Golgi-residential protein mediating ER-Golgi transport (in both anterograde and retrograde directions). Presynaptic precursors, established in the soma of neurons, are transported from the trans-Golgi network to synapses by Rab2. This process is essential for synapse formation and neurotransmission. It has also been shown that Rab2 functions upstream of Arl8 in the biogenesis of presynaptic neurons [103]. Rab7 is involved in tumor progression as it can be either an oncogenic or tumor suppressor, determined by environmental factors. It plays an essential role in chemoresistance to cisplatin by modulating the late endosomal pathway and lysosome biogenesis [104].

Data provided by this study suggests that Rab2 and Arl8 serve as potential targets for autophagy enhancement in the *Drosophila* nervous system. In the future, it might be interesting to assess the effect of Rab2 and Arl8 coactivation on autophagy, and it would also be worthwhile to validate these findings in a mammalian model and human cell lines. Molecules that specifically inhibit Rab2 or Arl8 serve as potent targets for drug candidates to modulate the activity of the autophagic process in treating neurodegenerative pathologies. In the future, it would be reasonable to investigate which GAP enzyme can inhibit Rab2 or Arl8 specifically, but not activate Rab7, with similar medical purposes.

## Figures and Tables

**Figure 1 cells-12-01753-f001:**
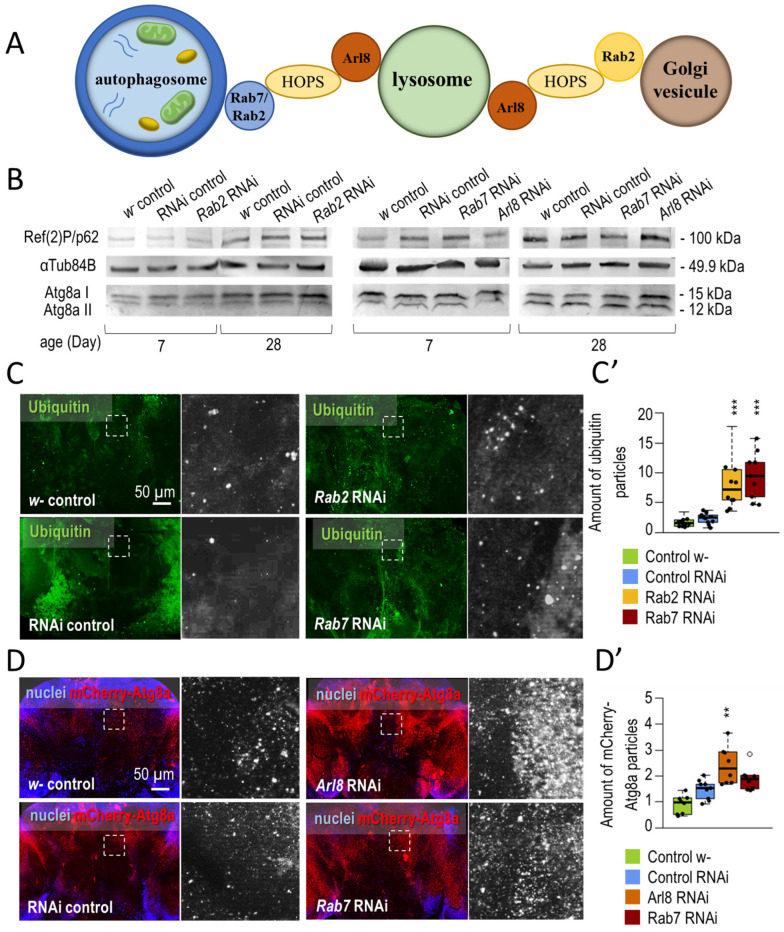
Inhibition of small GTPase proteins involved in controlling lysosomal degradation represses autophagy in neurons**.** (**A**) Roles of the small GTPase proteins Rab2, Rab7, and Arl8 during lysosome-dependent degradation pathways. (**B**) Western blot analysis shows autophagic activity in Rab2-, Arl8-, and Rab7-RNAi backgrounds. p62/Ref(2)P is a substrate of autophagy; thus, its levels inversely correlate with the autophagic flux. Atg8a has key roles during autophagy, as it regulates phagophore formation and closure, and is required for selective autophagy. Atg8a-I bands indicate soluble, while Atg8a-II bands represent membrane-bounded, form of the protein. αTub84B protein was used as an internal control. (**C**,**C’**) Ubiquitinated proteins and aggregates are digested with autophagy; therefore, the effect of small GTPase RNAi was also investigated with ubiquitin labelling (green). Silencing of Rab2 and Rab7 increases the amount of ubiquitinated aggregates (**D**,**D’**) The abundance of mCherry-Atg8a-positive (red) structures was observed in small GTPases RNAi background. The marker labels autophagic vesicles, forming phagophores, autophagosomes, amphisomes, and autolysosomes. Nuclei were stained by Hoechst dye (blue). Arl8 silencing increases the amount of mCherry-Atg8a structures. The white dashed line squares indicate the magnified detail in the fluorescence images. Boxes represent the most typical 50% of the samples, line indicates the median, and upper and lower whiskers show the remaining 25–25% of the samples. Circles mark outliers. **: *p* < 0.01, ***: *p* < 0.005; statistical analysis was performed as described in the Materials and Methods.

**Figure 2 cells-12-01753-f002:**
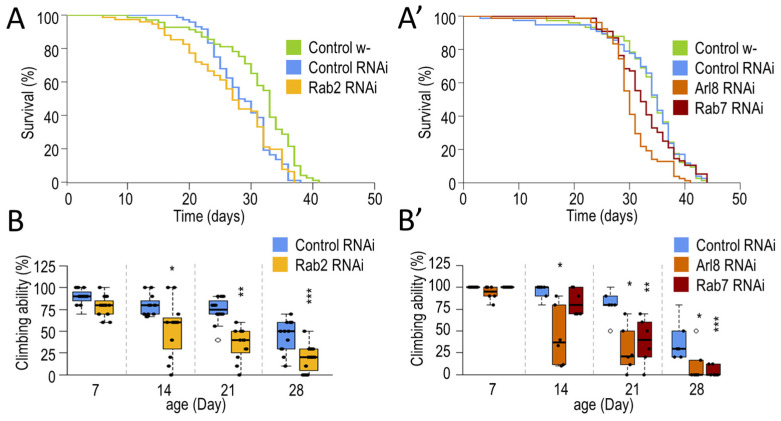
Neuronal silencing of small GTPases (Arl8, Rab2, Rab7) reduces lifespan and decreases climbing ability in *Drosophila*. (**A**,**A’**) Kaplan–Meier lifespan curves of *Drosophila*. Using a Ddc-Gal4 driver, Rab2, Arl8, and Rab7 were silenced only in dopaminergic and serotonergic neurons, and compared to isogenic *w^1118^* and ON-target free RNAi controls (both controls were crossed with Ddc-Gal4). Neuronal silencing of small GTPases reduces lifespan. (**B**,**B’**) Plots represent climbing ability of animals silenced for a small GTPase. Climbing was measured at 20 s. The lack of small GTPases impairs the climbing ability of animals. The boxes represent the most typical 50% of the samples, the line indicates the median, and upper and lower whiskers show the remaining 25–25% of the samples. Circles mark outliers. *: *p* < 0.05, **: *p* < 0.01, ***: *p* < 0.005; statistical analysis was performed as described in the Section 2.

**Figure 3 cells-12-01753-f003:**
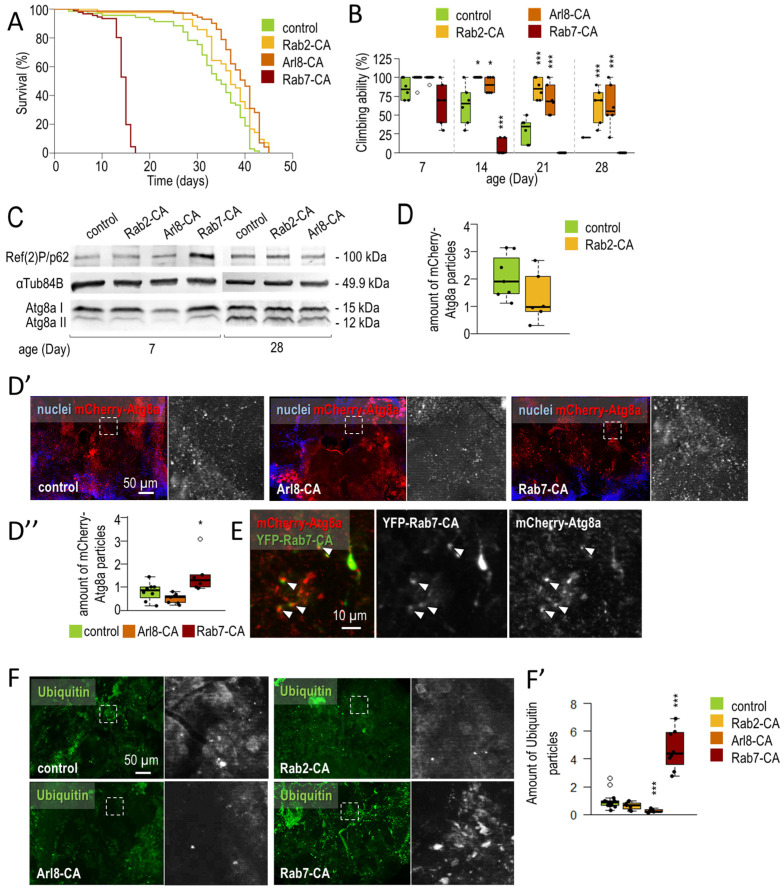
Neuron-specific activation of Rab2 and Arl8 extends lifespan and enhances autophagy. (**A**) Kaplan–Meier lifespan curves of *Drosophila*. Constitutively active forms of the small-GTPase proteins Rab2 (Rab2-CA), Arl8 (Arl8-CA), and Rab7 (Rab7-CA) in dopaminergic and serotonergic neurons were tested (using Ddc-Gal4 driver). Our results show that Rab2-CA and Arl8-CA increase, while Rab7-CA decreases lifespan. (**B**) Moving ability of Rab2-CA, Arl8-CA, and Rab-7-CA animals were observed at different adult ages. The activation of Rab2 and Arl8 oppositely varies the climbing ability of animals compared to Rab7 activation Climbing was measured at 20 s. (**C**) Western blot analysis showing p62/Ref(2) and Atg8a-II levels. Ref(2)P and Atg8a-II serve as substrates for autophagy; thus, their levels inversely correlate with autophagic capacity. αTub84B protein was used as internal control. For measurements, proteins were isolated from the head of young (7 days old) and old (28 days old) female flies. (**D**–**D’’**) The abundance of mCherry-Atg8a-positive (red) foci refers to autophagic vesicles, measured in small-GTPase activated and control genetic backgrounds. Nuclei were stained by Hoechst dye (blue). Rab7-CA increases the amount of mCherry-Atg8a structures, while Rab2 and Arl8 activation decreases them. (**E**) Colocalization error can be found between the colocalization of YFP-Rab7-CA-(YFP-bounded protein—green) and mCherry-Atg8a-positive (red) structures. (**F**,**F’**) Ubiquitinated aggregates are digested by autophagy. Amounts of ubiquitinated proteins (green) were measured in adult brains with hyperactivated small-GTPase proteins. Rab2 and Arl8 activation decreased, whereas Rab7 activation increased, ubiquitin aggregation in *Drosophila* brains. For the measurement, animals were maintained at 29 °C during adulthood, and Ddc-Gal4 triggered the overexpression of constitutively active form of small-GTPase. The white dashed line squares indicate the magnified detail in the fluorescence images. Boxes represent the most typical 50% of the samples, the line indicates the median, and upper and lower whiskers show the remaining 25–25% of the samples. Circles mark outliers. *: *p* < 0.05, ***: *p* < 0.005; statistical analysis was performed as described in the Materials and Methods.

**Figure 4 cells-12-01753-f004:**
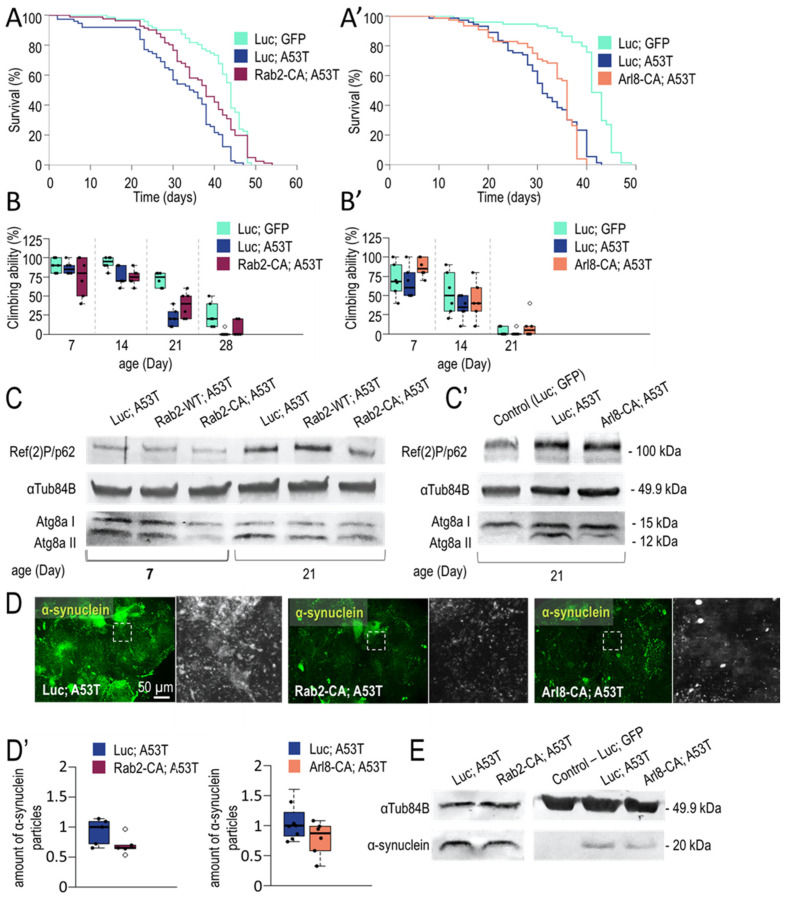
Rab2 hyperactivation promotes longevity and improves climbing ability in a Parkinson’s disease (PD) fly model. (**A**,**A’**) Kaplan–Meier graphs of *Drosophila* surviving ability in percentage. Rab2-CA and Arl8-CA were overexpressed in a background expressing the human A53T mutant α-synuclein, and compared with native (Luc; GFP) and disease (Luc; A53T) controls. Rab2 activation increased survival of A53T mutant animals (compared to disease control). (**B**,**B’**) Climbing ability was tested in PD model. Climbing was measured at 20, 40, and 60 s intervals, and changes are plotted on the diagrams (**B**) and (**B’**). Panel (**B**) shows climbing ability within 40 s, while diagram (**B’**) shows climbing ability within 20 s. The activation of Arl8 and Rab2 increases the climbing ability of A53T mutants. (**C**,**C’**) Autophagic activity was studied by Western blot. Atg8a-II and p62/Ref(2)P levels were identified. αTub84B was used as an internal control. (**D**,**D’**) Human mutant (A53T) α-synuclein (green) was stained by immunohistochemistry and observed by a fluorescent microscope. Arl8-CA and Rab2-CA increase the degradation of mutant α-synuclein protein. (**E**) A53T level in PD models was also measured by Western blotting. αTub84B was used as an internal control. Animals were maintained at 29 °C during the adulthood, and the constitutively active form of small-GTPase were triggered by Ddc-Gal4 driver. The white dashed line squares indicate the magnified detail in the fluorescence images. Boxes represent the most typical 50% of the samples, the line indicates the median, and upper and lower whiskers show the remaining 25–25% of the samples. Circles mark outliers. statistical analysis was performed as described in Section 2.

## Data Availability

The datasets used and/or analyzed during the current study available from the corresponding author on reasonable request.

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
