# Peer review of "Potent New Targets for Autophagy Enhancement to Delay Neuronal Ageing"

_cells, 2023, doi:10.3390/cells12131753_

Round 1
Reviewer 1 Report
The article investigates an always-pressing topic of neuronal aging. In a series of convincing experiments, the authors discover the mechanistic importance of small GTPases in regulating autophagy not only in the context of neuronal aging but also in the background of neurodegenerative diseases such as Parkinson's. We think this work has an important weight in autophagy research and incites the motivation to conduct further experiments on the mentioned autophagy regulators in a mammal or human backgrounds. We fully support the publishing of this work with some suggestions for correction/ clarification mentioned in the attached review.

Author Response
Dear Reviewer,
thank you very much for your constructive remarks. We believe that our manuscript has improved a lot after the corrections. We have tried to answer all your requests and questions. Please find below our response to each of your questions in italics.
General comments:
- Introduction: Line 11- 13; this sentence seems to be an instruction of the journal on how to format the Abstract. Did the authors forget to delete this? Although the introduction is already quite extensive, the authors could review also the general role of autophagy in aging and the current status of non-genetical interventions, such as the effects of anti-aging substances (such as polyamines, etc.) on autophagy maintenance in aging.
Thank you for the relevant request. We have included the Introduction chapter with the requested points. All changes are marked with a proofreading mark.
- Materials and methods: in the Climbing Assay description you refer to the original method description as in (31). After looking into the source I´m still missing a bit more detailed description; for example; did you separate females and males and how much before the assay? And you use only female flies in your results? 1h recovery period seems a bit too short to properly recover. How did you conduct this protocol, is there an original Negative Geotaxis protocol that you adapted from?
Thank you for the relevant question. The missing data have been added to the Materials and Methods chapter: in the climbing tests, we have examined the sexes separately because there could bea significant difference between the climbing ability of males and females. We tested 10 animals in a single tube, with several parallel experiments per genotype. The average climbing ability of each point on the box-plots is shown for these parallel measurements.
We examined both females and males in our tests, with the changes in the two sexes being in the same direction. Since only female samples were used for the cell biology and Western blot experiments, we have included only the female data in the manuscript for the climbing tests. 1.5 h recovery time was given to the animals after anesthesia. This was corrected in the Materials and Methods section. Based on our previous experience, this time is sufficient for the regeneration of Drosophila. No significant improvement in climbing ability was observed when animals were allowed more than 1.5 hours between anesthesia and climbing.
- In the Western Blotting description, I´m missing some information; does a 15 microliter fly head sample come from 1 fly head homogenate? What lysis buffer do you use?
The missing data have been added to the Materials and Methods chapter: For protein samples, solutions containing 2 mg protein were prepared using 15 female heads. The heads were homogenized in 48ul lysis buffer and 48ul mercaptoethanol-containing lysis buffer (BioRad 1610737). The lysis buffer contained the following components: 0.5% Tween 20, 25 mM Tris pH 7.5, 150 mM NaCl, 2 mM EDTA, 20 mM Nicotinamide, and 6.51 ml MQ water.
- PCR and quantitative PCR description; also here I´d like to know, did you use only 1 female fly head? Or how many?
Thank you for your relevant comment. We have added the requested information to the Materials and Methods chapter: For RNA isolation, 10 female heads were used for each sample.
Major:
- Figure 1: B) How many repetitions were done for Western Blots? Would it be possible to quantify the band intensities? Tubulin doesn´t look very intact across the lanes but considers this as a minor observation. In the description of the figure, line 377 índuces inhibits´. You also refer to the 20-second C and C´diagram. Where are those?
Western blot experiments were repeated at least three times. We performed the evaluation of the Western blot results using Image J software, data are presented in Supplementary Figures.
- Figure 2: Line 395; ´motility´ is a different thing than climbing.
Motility and similar terms have been replaced in the text by climbing.
Minor:
- Figure 1: D) for this picture (and the following such pictures within the article) I would use a different color for the heading, or make a legend next to it or write above the picture. It gets a bit difficult to read on this background.
We have made the fluorescent picture's colour captions more visible by adding a background, new colours and shading. Furthermore, we have also included enlarged details of the same regions of the brain in the figures.
- Figure 3: line 407 áctivation´, would rather use the term overexpression´since it is an overexpression experiment and not just activation.
Thank you for your suggestion. We have amended the text to read: Overexpression of the activated forms of Rab2 and Arl8 extends lifespan.
- The authors sometimes forgot the brackets when citing references in the text, for example, such as in line 77 and line 494.
Thank you for your suggestion, we have tried to correct the spelling and grammatical errors throughout the paper.
- I would suggest avoiding broad terms to describe the results, such as in line 313, where ´physiology' could mean much more than evidently meant locomotion ability.
Thank you for your comments. We have corrected the error in the text.
- Also, what is noticed, within the text you use 3 different terms to refer to locomotion ability. Please unify the terminology.
We have standardized the terminology for climbing ability throughout the manuscript.
- Please spell-check for commas and proper formatting of sentences.
Thank you for your suggestion, we have tried to correct the spelling and grammatical errors throughout the paper.

Reviewer 2 Report
The authors Szinyákovics et al., have submitted an original research article entitled "Potent new targets for autophagy enhancement to delay neuronal aging" where they discuss the effect of regulating autophagy through modulating the levels of autophagy regulators like Rab7, Rab2, and Arl8. While the premise of the article is good, the authors need to work on the manuscript extensively to make it scientifically robust. A few major comments on the manuscript are:
In figure S1, the RNAi results do not indicate a significant decrease in the levels of Rab7.
In figure 1B, the authors state that there is an increase in the levels of p62 in all 3 RNAi-treated flies as compared to the controls. However, the levels of tubulin are correspondingly higher in those RNAi-treated samples. Therefore, it would be helpful if the authors could quantify all their western blots and revise the figure to include a representative western blot with the quantification. Also, describing these results in an age-dependent manner will help to increase the significance of these results. Further, the authors haven't discussed the results of Atg8a and their significance concerning the results of p62.
In Figure 1C, the expression of ubiquitin is significantly lower only in Arl8 RNAi flies. No such effect has been observed with Rab7 RNAi flies. Also, the authors have not included the results of Rab2 RNAi flies. Therefore, the authors need to revise the results accordingly.
In Figure 2A and S2A, the authors state that there is a significant decrease in the survival rate of flies expressing Rab2 RNAi. However, there is a significant drop in the survival rate of the flies expressing control RNAi as well. Therefore, a decrease in survival of Rab2 RNAi expressing flies could be a combined effect of survival decrease due to control RNAi and Rab2 RNAi. The authors need to mention this possibility in their results.
Also, in Figure 2A' and S2A', the level of survival of control flies appears to be significantly lower than the survival of the same flies in the experiments in Figure 2A, and S2A. In this context, the robustness of the entire set of experiments cannot be relied upon to provide accurate results for this parameter.
In figures 2B, B', the authors state that there is an impairment of the animals' climbing ability for flies expressing RNAi against either of the three genes. However, the results do not appear to be significantly different from the control for Rab2 and for other genes, the effect shows up from 14 days for Arl8 RNAi and 21 days for Rab7 RNAi. The authors should give a more detailed description of these results.
Comparing Figures 3B and S3B, the levels of increase in the climbing ability of the Rab2-CA flies as compared to control is significantly different in both sets of experiments. The authors should address this discrepancy. Also, the authors have not explained the reason for trying Rab2-WT over-expression vs Rab2-CA and why they did not do such analyses for other genes.
In figures S3C, and C' the authors have shown a difference in expression levels of Rab2-CA and Rab7-CA through fluorescence microscopy. While fluorescence intensity does provide an analysis of expression levels, a more robust assay would be western blotting. Since the authors may already have the samples from Figure 3C, it would be helpful if the authors could include western blot analysis.
In Figure S3D, the decrease in climbing ability in Rab7-CA #9779 and Rab7-CA #50785 expressing flies does not appear to be significant. Therefore, the data is not in coherence with the point the authors are trying to make.
Author Response
Dear Reviewer,
thank you very much for your constructive remarks and questions. We have corrected many errors in the manuscript based on your feedback. We have also been able to answer several relevant questions. Overall, we believe that our manuscript has improved significantly during the revision process and we are optimistic that it will be favorably received.
Please find the answers to your questions in the attached file.

Round 2
Reviewer 2 Report
The authors Szinyákovics et al., have submitted a revised manuscript entitled "Potent new targets for autophagy enhancement to delay neuronal aging". The revised manuscript has been improved on all the lines indicated during the previous round of revision. A few minor areas of concern remain:
The authors have repeated the term autophagosome twice on lines 63 and 64. One of them should be removed.
In the introduction section, the authors have described the general role of autophagy in neurological diseases. It is better to cite recent reviews for this. Instead, the introduction section could focus on what is already known about Rab2, Rab7, and Arl8-mediated regulation of autophagy and its effects on neurological diseases in different model systems.
The authors must cite the statements at the start of each sub-section in the Results section.
In Figure S1A, the authors should depict the changes in mRNA abundance, measured through qPCR, normalized to Actin. The authors could refer to https://pubmed.ncbi.nlm.nih.gov/11846609/ for details on how to do these calculations. Also, it would be helpful if the authors demonstrated Rab2 knockdown at both transcript and protein levels similar to Rab7 and Arl8.
In section 3.1.1., the authors should cite studies showing the importance of autolysosome formation or small GTPases in regulating Drosophila's physiological and cognitive abilities.
Author Response
Dear Reviewer!
We are delighted to read that you are satisfied with our work on the manuscript during the revision. Please find the attached pdf file of our response point-by-point to your requests during the minor revision. All new changes have been proofread in the manuscript.
We believe that we have also managed to improve our manuscript during the minor revision. We are optimistic that the new version will also be well received.
